# A Max-Min Entropy Framework for Reinforcement Learning

**Seungyul Han**
Graduate School of Artificial Intelligence
UNIST
Ulsan, South Korea 44919
syhan@unist.ac.kr

**Youngchul Sung**[†]
School of Electrical Engineering
KAIST
Daejeon, South Korea 34141
ycsung@kaist.ac.kr

## Abstract

In this paper, we propose a max-min entropy framework for reinforcement learning (RL) to overcome the limitation of the soft actor-critic (SAC) algorithm implementing the maximum entropy RL in model-free sample-based learning. Whereas the maximum entropy RL guides learning for policies to reach states with high entropy in the future, the proposed max-min entropy framework aims to learn to visit states with low entropy and maximize the entropy of these low-entropy states to promote better exploration. For general Markov decision processes (MDPs), an efficient algorithm is constructed under the proposed max-min entropy framework based on disentanglement of exploration and exploitation. Numerical results show that the proposed algorithm yields drastic performance improvement over the current state-of-the-art RL algorithms.

## 1 Introduction

The maximum entropy framework has been considered in various RL domains [22, 23, 30, 45, 51, 53, 58]. Maximum entropy RL adds the expected policy entropy to the return objective of standard RL in order to maximize both the return and the entropy of policy distribution. Maximum entropy RL encourages the policy to choose multiple actions probabilistically and yields a significant improvement in exploration and robustness and good final performance in various control tasks [15, 20, 21, 25, 26, 29, 50]. In particular, soft actor-critic (SAC) implements maximum entropy RL in an efficient iterative manner based on soft policy iteration and guarantees convergence to the optimal policy for finite MDPs, yielding significant performance improvement over various on-policy and off-policy recent RL algorithms in many continuous control tasks. However, we observe that such an iterative implementation of the maximum entropy strategy of optimizing for policies that aim to reach states with high entropy in the future does not necessarily result in the desired exploration behavior but may yield positive feedback hindering exploration in model-free sample-based learning with function approximation. In order to overcome such limitations associated with implementation of the maximum entropy RL, we propose a max-min entropy framework for RL, which aims to learn policies reaching states with low entropy and maximizing the entropy of these low-entropy states, whereas the conventional maximum entropy RL optimizes for policies that aim to visit states with high entropy and maximize the entropy of those high-entropy states for high entropy of the entire trajectory. We implemented the proposed max-min entropy framework into a practical iterative actor-critic algorithm based on policy iteration with disentangled exploration and exploitation. It is demonstrated that the proposed algorithm significantly enhances exploration capability due to the fairness across states induced by the max-min framework and yields drastic performance improvement over existing RL algorithms including maximum-entropy SAC on difficult control tasks.

35th Conference on Neural Information Processing Systems (NeurIPS 2021).

## 2   Related Works

**Maximum Entropy RL**: The maximum entropy framework has been considered in various RL domains: inverse reinforcement learning [58], stochastic optimal control [45, 51, 53], guided policy search [30], and off-policy learning [22, 23]. There is a connection between value-based and policy-based RL under the policy entropy regularization [38], [42] combines them, and finally [46] proves that they are equivalent. Maximum entropy RL is also related to probabilistic inference [40, 45]. Recently, maximizing the entropy of state distribution instead of the policy distribution [26] and maximizing the entropy considering the previous sample action distribution [25] have been investigated for better exploration.

**Max-Min Optimization**: Max-min optimization aims to maximize the minimum of the objective function [11]. Under the convex-concave assumption, there exist many algorithms to find the solution to a max-min problem by using optimistic mirror descent [44], Frank-Wolfe algorithm [17], and Primal-Dual method [24]. However, non-convex max-min problems are more challenging [37] and there are several recent studies to find (approximate) solutions to non-convex max-min optimization problems [6, 41, 43]. This framework has been used in various optimization/control domains: fair resource allocation [31], inference [4, 56], generative adversarial network (GAN) [2, 18], robust training [33], and reinforcement learning [54].

**Exploration in RL**: Exploration is one of the most important issues in model-free RL, as there is the key assumption that all state-action pairs must be visited infinitely often to guarantee the convergence of $Q$-function [55]. In order to explore diverse state-action pairs in the joint state-action space, various methods have been considered in prior works: intrinsically-motivated reward based on curiosity [5, 10], model prediction error [1, 9], information gain [25, 27, 28], and counting states [32, 34]. These exploration techniques improve exploration and performance in challenging sparse-reward environments [3, 9, 12].

## 3   Background

### 3.1   Basic RL Setup

We consider an infinite-horizon MDP $(\mathcal{S}, \mathcal{A}, P, \gamma, r)$, where $\mathcal{S}$ is the state space, $\mathcal{A}$ is the action space, $P$ is the transition probability, $\gamma$ is the discount factor, and $r$ is the bounded reward function. We assume that each action dimension is bounded. The RL agent has a policy $\pi : \mathcal{S} \times \mathcal{A} \to \mathbb{R}^+ \in \Pi$, which chooses an action $a_t$ for given state $s_t$ according to $a_t \sim \pi(\cdot|s_t)$ at each time step $t$, where $\Pi$ is the policy space. For action $a_t$, the environment yields the reward $r_t := r(s_t, a_t)$ and the next state $s_{t+1} \sim P(s_{t+1}|s_t, a_t)$. Standard RL learns policy $\pi$ to maximize the discounted return $\mathbb{E}_{s_0 \sim p_0, \tau_0 \sim \pi}[\sum_{t=0}^{\infty} \gamma^t r_t]$, where $\tau_t = (s_t, a_t, s_{t+1}, a_{t+1}, \cdots)$ is an episode trajectory.

### 3.2   Maximum Entropy RL and Soft Actor-Critic

Maximum entropy RL maximizes both the expected return and the expected policy entropy simultaneously to achieve an improvement in exploration and robustness. The entropy-augmented objective function of maximum entropy RL is given by

$$J_{MaxEnt}(\pi) = \mathbb{E}_{s_0 \sim p_0, \tau_0 \sim \pi} \left[ \sum_{t=0}^{\infty} \gamma^t (r_t + \alpha \mathcal{H}(\pi(\cdot|s_t))) \right], \tag{1}$$

where $\mathcal{H}(\pi(\cdot|s)) = \mathbb{E}_{a \sim \pi(\cdot|s)}[-\log \pi(a|s)]$ is the entropy function and $\alpha > 0$ is the entropy coefficient. A key point here is that the policy entropy is included in the reward not used as an external regularizer at each time step. Thus, *this maximum entropy RL framework optimizes for policies that aim to reach states on which policies have high entropy in the future* [21].

Soft actor-critic (SAC) is an efficient off-policy actor-critic algorithm to solve the maximum entropy RL problem [22]. SAC maximizes (1) based on soft policy iteration, which consists of soft policy evaluation and soft policy improvement. For this, the soft $Q$-value of given $(s_t, a_t)$ is defined as

$$Q^\pi(s_t, a_t) := r_t + \mathbb{E}_{\tau_{t+1} \sim \pi} \left[ \sum_{l=t+1}^{\infty} \gamma^{l-t} (r_l + \alpha \mathcal{H}(\pi(\cdot|s_l))) \right], \tag{2}$$

which does not include the policy entropy of the current time step but includes the sum of all future policy entropy and the sum of all current and future rewards. For given $\pi$, soft policy evaluation guarantees the convergence of soft $Q$-function estimation, which estimates $Q^\pi$ by iteratively applying a modified Bellman operator $\mathcal{T}^\pi$ to a real-valued estimate function $Q : \mathcal{S} \times \mathcal{A} \to \mathbb{R}$, given by

$$\mathcal{T}^\pi Q(s_t, a_t) = r_t + \gamma \mathbb{E}_{s_{t+1} \sim P(\cdot|s_t, a_t)}[V(s_{t+1})], \quad \text{where} \tag{3}$$
$$V(s_t) = \mathbb{E}_{a_t \sim \pi(\cdot|s_t)}[Q(s_t, a_t) - \alpha \log \pi(a_t|s_t)]$$

and $V(s_t)$ is the soft state value function. Then, at each iteration, SAC updates the policy as

$$\pi_{new} = \underset{\pi \in \Pi}{\arg \min} \, D_{KL}\left(\pi(\cdot|s_t) \middle\| \frac{\exp(Q^{\pi_{old}}(s_t, a_t)/\alpha)}{Z^{\pi_{old}}(s_t)}\right) \tag{4}$$
$$= \underset{\pi \in \Pi}{\arg \max} \, \mathbb{E}_{a_t \sim \pi(\cdot|s_t)}[Q^{\pi_{old}}(s_t, a_t) - \alpha \log \pi(a_t|s_t)] \tag{5}$$

where $Z^{\pi_{old}}(s_t)$ is the log partition function which is a function of $s_t$ only. Soft policy improvement guarantees $Q^{\pi_{new}}(s_t, a_t) \geq Q^{\pi_{old}}(s_t, a_t)$ for all $(s_t, a_t) \in \mathcal{S} \times \mathcal{A}$. Finally, soft policy evaluation and soft policy improvement are repeated. Then, any initial policy $\pi \in \Pi$ converges to the optimal policy $\pi^*$, i.e., $Q^{\pi^*}(s_t, a_t) \geq Q^{\pi'}(s_t, a_t)$ for all $\pi' \in \Pi$ and all $(s_t, a_t) \in \mathcal{S} \times \mathcal{A}$, and $\pi^*$ maximizes $J_{MaxEnt}$ [22]. Proof of soft policy iteration assumes finite MDPs. SAC approximates the soft policy iteration by sample-based learning with function approximation in continuous-space cases.

## 4 Motivation: Limitation of Maximum Entropy SAC in Pure Exploration

In this section, we will consider only the maximum entropy SAC in a pure exploration setup without the reward function (the reward function $r = 0$ in MDPs). As seen in Sec. 3, SAC efficiently solves the maximum entropy RL problem to maximize (1) in an iterative manner based on judiciously-defined state and action value functions and the step-wise optimization cost (5). The well-defined value functions and the local cost function as such enable proof of soft policy improvement for finite MDPs in a similar way to the proof of the classical policy improvement theorem. Note that at each time step, SAC updates the policy to maximize the cost (5), composed of two terms: $\mathbb{E}_{a_t \sim \pi(\cdot|s_t)}[Q^{\pi_{old}}(s_t, a_t)]$ and $\alpha \mathbb{E}_{a_t \sim \pi(\cdot|s_t)}[-\log \pi(a_t|s_t)] = \alpha \mathcal{H}(\pi(a_t|s_t))$. As aforementioned, the soft $Q$-function contains the sum of current and future rewards and the sum of only future policy entropy. Since we consider only the entropy terms without rewards here, the first term $\mathbb{E}_{a_t \sim \pi(\cdot|s_t)}[Q^{\pi_{old}}(s_t, a_t)]$ is the current estimate of the sum of future entropy when action $a_t$ is taken from policy $\pi$ at state $s_t$, whereas the second term $\alpha \mathcal{H}(\pi(a_t|s_t))$ is the entropy of the policy $\pi$ itself. Hence, at each time step, SAC tries to update the policy $\pi$ to yield the maximum sum of the estimated future entropy and the policy entropy itself. Here, the term $\mathbb{E}_{a_t \sim \pi(\cdot|s_t)}[Q^{\pi_{old}}(s_t, a_t)]$ plays the role of guiding the policy towards the direction of high future entropy.

**Saturation:** In sample-based update with function approximation, however, the SAC iteration does not yield the desired result, contrary to the intention behind maximum entropy. To see this, let us consider a pure exploration task in which there is no reward. The considered task is a $100 \times 100$ continuous 4-room maze proposed in [25], modified from the continuous grid map available at `https://github.com/huyaoyu/GridMap`. Fig. 1(a) shows the maze environment, where state is the $(x, y)$-position of the agent in the map, action is $(dx, dy)$ bounded by $[-1, 1] \times [-1, 1]$, and the next state of the agent is $(x + dx, y + dy)$. Starting from the left-lower corner $(0.5, 0.5)$, the agent explores the maze without any external reward. First, note that for this pure exploration task, the optimal policy maximizing $J_{MaxEnt}(\pi)$ is given by the uniform policy that selects all actions in $\mathcal{A} = [-1, 1] \times [-1, 1]$ uniformly regardless of the value of $s_t$. This is because the uniform distribution has maximum entropy for a bounded space [13]. Then, we compare the exploration behaviour of SAC and the uniform policy in the maze task. Fig. 1(b) shows the mean accumulated number of different visited states averaged over 30 random seeds as time goes, where the shaded region in the curve represents standard deviation ($1\sigma$) from the mean and a different state is meant as a nonoverlapping quantized $1 \times 1$ square. As seen in Fig. 1(b), SAC explores more states than the uniform policy at the early stage of learning. As learning progresses, however, SAC fails to visit new states after $300k$ time steps, whereas the uniform policy continues visiting new states. As a result, SAC eventually visits fewer states than the uniform policy on average. The result shows that SAC fails to converge to the optimal uniform policy and its performance become saturated.

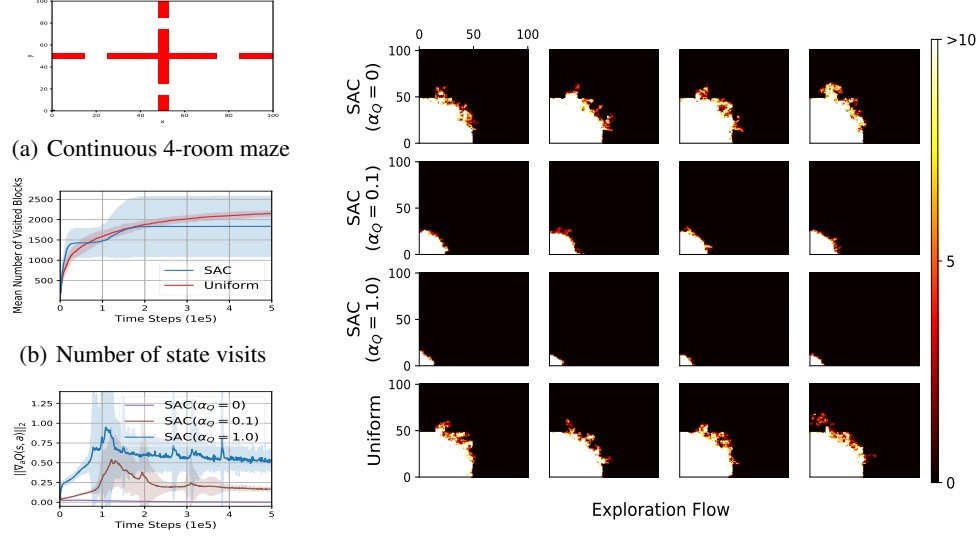

(a) Continuous 4-room maze

(b) Number of state visits

(c) Gradient of $Q$-function

(d) State histogram over 50k time steps starting from 300k, 350k, 400k, and 450k time steps (from the left in order)

Figure 1: Comparison of SAC and the uniform policy in the continuous 4-room maze

**Narrow Exploration Radius:** To examine the saturation behavior of SAC in the above pure exploration task, we investigate the policy update of SAC in (5). Since the current $Q$-function estimate (implemented by a neural network) replaces $Q^{\pi_{old}}$ in (5) in implementation with function approximation, the policy update is rewritten as

$$\arg\max_{\pi \in \Pi}\{\mathbb{E}_{a_t \sim \pi(\cdot|s_t)}[Q(s_t, a_t)] + \alpha\mathcal{H}(\pi(\cdot|s_t))\}. \tag{6}$$

As mentioned already, the first term $\mathbb{E}_{a_t \sim \pi(\cdot|s_t)}[Q(s_t, a_t)]$ is the current estimate of the sum of future entropy in this pure exploration case when action $a_t$ is taken from policy $\pi$ at state $s_t$, whereas the second term $\alpha\mathcal{H}(\pi(a_t|s_t))$ is the entropy of the policy $\pi$ itself. The first term $\mathbb{E}_{a_t \sim \pi(\cdot|s_t)}[Q(s_t, a_t)]$ intends to direct the policy towards the direction of high future entropy. Note that maximizing the second term already yields the uniform policy, but the $Q$-function term affects the policy update. In order to see how the $Q$-function term actually affects the policy update, we differentiate the entropy coefficient $\alpha$ in the policy update part (5) or (6) as the policy entropy coefficient $\alpha_\pi$ and that in the soft value function part (2) and (3) as the value entropy coefficient $\alpha_Q$. We fix $\alpha_\pi$ as $\alpha_\pi = 1$ and change $\alpha_Q$ as 0, 0.1, and 1 (note that the case of $\alpha_Q = 1$ is original SAC). With this change of $\alpha_Q$, we conducted the same pure exploration maze task. Fig. 1(c) shows the average norm of the gradient of $Q$-function with respect to action, i.e., $\mathbb{E}_{s_t \sim \mathcal{D}}[||\nabla_a Q(s_t, a)|_{a=a_t}||]$ over time with $a_t \sim \pi(\cdot|s_t)$ and $s_t$ from a mini-batch drawn from the replay buffer $\mathcal{D}$ of SAC update, where the $Q$ neural network weights were initialized randomly. Fig. 1(d) shows the histogram of states that the policy visits over 50k time steps starting from 300k, 350k, 400k, and 450k time steps. When $\alpha_Q = 0$ with no reward, the $Q$-function update by the Bellman operator $\mathcal{T}^\pi$ in (3) is trivial as $Q(s, a) \leftarrow \mathbb{E}_{s' \sim P(\cdot|s,a), a' \sim \pi(\cdot|s')}[Q(s', a')]$, i.e., replacement. When the initial $Q(s, a)$ is (nearly) flat over $\mathcal{S} \times \mathcal{A}$ by initial random weight assignment for the $Q$-neural network, the flat $Q$ is maintained by this trivial update. Indeed, it is seen in Fig. 1(c) that $\mathbb{E}_{s_t \sim \mathcal{D}}[||\nabla_a Q(s_t, a)|_{a=a_t}||]$ with $a_t \sim \pi(\cdot|s_t)$ is nearly zero across all time for $\alpha_Q = 0$. With a flat function $Q(s_t, \cdot) \approx c$ over the action space $\mathcal{A}$, the first term $\mathbb{E}_{a_t \sim \pi(\cdot|s_t)}[Q(s_t, a_t)]$ in (6) does not affect the policy update, only the second term $\mathcal{H}(\pi(\cdot|s_t))$ works, and thus the policy update yields $\pi$ to converge to the uniform policy for every state maximizing the total entropy. Hence, the exploration radius in the case of $\alpha_Q = 0$ is almost the same as that of the uniform policy, as seen in Fig. 1(d). When $\alpha_Q > 0$, on the other hand, the $Q$-function starts to be updated nontrivially by the Bellman operator $\mathcal{T}^\pi$ in (3) due to the $-\log \pi(a_{t+1}|s_{t+1})$ term in $V(s_{t+1})$ in (3), with $\pi$ given by the current policy. It is now seen in Fig. 1(c) that $\mathbb{E}_{s_t \sim \mathcal{D}}[||\nabla_a Q(s_t, a)|_{a=a_t}||]$ is not zero anymore and the gradient norm becomes larger as $\alpha_Q$ increases from 0.1 to 1.0. Non-zero $\mathbb{E}_{s_t \sim \mathcal{D}}[||\nabla_a Q(s_t, a)|_{a=a_t}||]$ means that $Q(s_t, \cdot)$ as a function of action $a_t$ for given $s_t$ is not flat anymore and the first term in (6) affects the policy update so that the policy is updated for the direction of high $Q$-value (with intention for high future entropy) as well

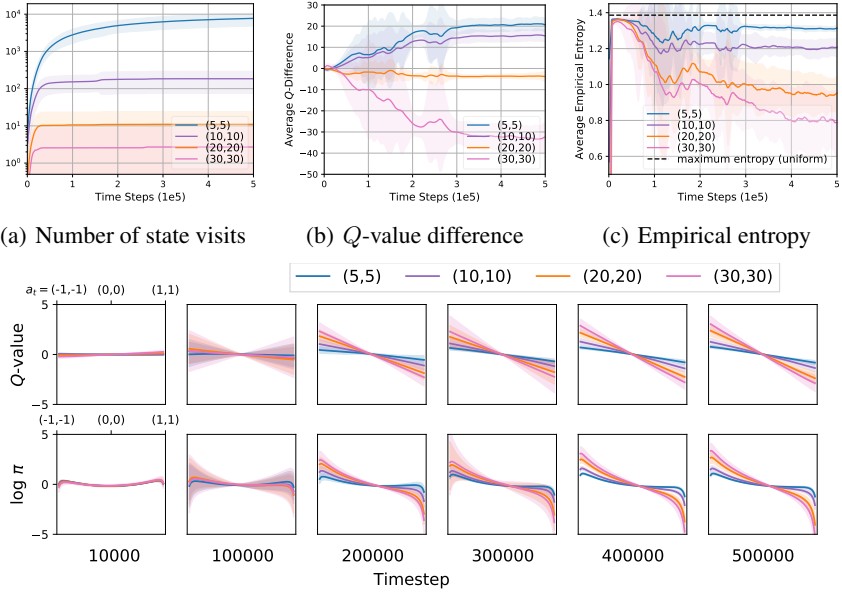

(a) Number of state visits     (b) $Q$-value difference     (c) Empirical entropy

(d) Cross-section of $Q$-function and $\log \pi$ along the action line at square center

Figure 2: Analysis on sample trajectories of SAC in the continuous maze task

as high policy entropy $\mathcal{H}(\pi)$. As seen in Fig. 1(d), however, the exploration radius reduces as $\alpha_Q$ increases from 0 to 1. The iteration process does not evolves for wider exploration as intended.

## 5 Methodology

### 5.1 A Deeper Look at Pure Exploration

In order to propose our new approach overcoming the limitation of SAC implementation of the maximum entropy framework, we first take a deeper look at how the $Q$-function term in (6) hinders exploration, as SAC (with $\alpha_\pi = \alpha_Q = 1$) learns the maze task. For this, we consider four $2 \times 2$ squares centered at $(5, 5)$, $(10, 10)$, $(20, 20)$, and $(30, 30)$ in the $100 \times 100$ maze, where every episode starts from (0.5,0.5). Fig. 2(a) shows the number of accumulated visits to each square as time elapses. Figs. 2(b) and 2(c) show the estimated $Q$ value and the average empirical entropy of each square, respectively, as time goes. For Fig. 2(b), every 1000 time steps, we sampled 1000 states uniformly from each square and an action from the current policy for each sampled state, and computed the $Q$-value average over the 1000 state-action samples for each square. Then, we computed the mean value of the four average values of the four squares. Fig. 2(b) shows the average Q value of each square relative to the four-square mean value as time goes. For Fig 2(c), every 1000 time steps, we sampled 1000 states uniformly from each square and computed the average empirical entropy $\mathbb{E}_{s_t}[\mathbb{E}_{a_t \sim \pi(\cdot|s_t)}[-\log \pi(a_t|s_t)]]$ of the current policy $\pi$ at time $t$ averaged over the 1000 sampled states $\{s_t\}$ from each square. The upper row of Fig. 2(d) shows the cross-section of the estimate $Q$-function $Q(s, a)$ along the diagonal action line from $(-1, -1)$ to $(1, 1)$ at the center state $s$ of each square, as time goes, where each curve is shifted in $y$-axis so that the mean value averaged over samples along the action line is matched to zero in $y$-axis. The lower row of Fig. 2(d) shows the value of $\log \pi(a|s)$ of the current policy $\pi$ at time step $t$ along the diagonal action line from $(-1, -1)$ to $(1, 1)$ at the center state $s$ for each square as time goes, where the curve is shifted in $y$-axis to match the mean value to zero in $y$-axis.

First, note from Fig. 2(a) that the farther a state is from the starting point $(0.5, 0.5)$, the less the agent visits the state, and the visitation difference is large. At the early stage of learning starting with random $Q$-network weight initialization and random policy-network weight initialization, there is little $Q$-value difference with respect to either state or action, as seen in Figs. 2(b) and 2(d), so the entropy term is dominant in the policy update (6) and the policy entropy increases with the policy distribution approaching the uniform distribution, as seen in Fig. 2(c). As time goes, learning of

the $Q$-function with the Bellman backup (3) progresses. Basically, the Bellman backup (3) with no reward adds $\Delta Q_t = \gamma\{\mathbb{E}_{a_{t+1}\sim\pi(\cdot|s_{t+1})}[Q(s_{t+1}, a_{t+1})] + \alpha\mathcal{H}(\pi(\cdot|s_{t+1}))\} - Q(s_t, a_t)$ to $Q(s_t, a_t)$ for every $(s_t, a_t) \in \mathcal{S} \times \mathcal{A}$. However, this is approximated in practical RL. In sample-based off-policy learning with function approximation, RL typically stores visited state-action pairs in the replay buffer $\mathcal{D}$ and the above Bellman backup is approximated as updating the $Q$-function by minimizing the loss $\mathbb{E}_{(s_t,a_t)\sim\mathcal{D}}[(Q(s_t, a_t) - Q^{target}(s_t, a_t))^2]$ based on a mini-batch uniformly drawn from the buffer. Under this off-policy learning with experience replay, when the initial $Q$-function is roughly flat and small, $\Delta Q_t$ soon becomes positive (the policy update increases the entropy of the visited states and $\Delta Q_t$ soon becomes positive for the visited states), and hence the $Q$-values of frequently-visited states are updated more and thus have higher $Q$-values as seen in Fig.2(b). This is because these states are stored more into $\mathcal{D}$ and sampled more from $\mathcal{D}$ at mini-batch generation. Then, the initial $Q$-value difference biases the policy to visit the states with high $Q$-values more frequently than the states with low $Q$-values because the policy is updated to choose actions that maximizes the expectation of $Q$-value $\mathbb{E}_{a_t\sim\pi(\cdot|s_t)}[Q(s_t, a_t)]$ in the policy update (6). This is evident in the Fig. 2(d), which shows the estimate of $Q$-function and the value of $\log\pi$ along the diagonal line. At the early stage of learning (10k time step in the figure), $Q$-function is roughly flat and the policy is almost close to the uniform distribution for the action line. As the time steps go on, the $Q$-values of actions close to $(-1, -1)$ becomes higher than the $Q$-value of actions near $(1, 1)$ due to the off-policy learning with experience replay, as explained above. Then, the policy is updated to choose actions with high $Q$-values more frequently to maximize the $Q$-value expectation, so the probability of choosing action $(-1, -1)$ towards the left-lower corner becomes higher than that of action $(1, 1)$ for the opposite direction. As the policy distribution leans toward a certain action and becomes asymmetric away from uniformity, the policy entropy decreases further. As seen in Fig. 2(c), the speed of the policy entropy decrease varies depending on the $Q$-value difference along the action line in Fig. 2(d), and the policy entropy difference deepens the $Q$-value difference between states in Fig. 2(b) because the $Q$-value estimates the policy entropy sum of future states. This positive feedback continues until saturation, as seen in Fig. 2(b), and it results in the narrow exploration radius in Fig. 1(d) because the policy will be forced to visit states with high $Q$-values only. Note that this positive feedback reduces the policy entropy due to the $Q$-value difference, contrary to the intention behind maximum entropy.

## 5.2 Max-Min Entropy RL

In order to break the unwanted positive feedback loop occurring when implementing the maximum entropy framework (i.e., max-max entropy framework) in the previous subsection, we must reduce the policy entropy difference between states to reduce the $Q$-value difference between states in the feedback loop. For this, we aim to learn the $Q$-function so that the policy visits states with low entropy, and the policy update increases the policy entropy of these low-entropy states. Under this principle, we propose a new max-min entropy (MME) framework that aims to learn the $Q$-function to estimate the negative sum of policy entropy, while maintaining the policy entropy maximization term $\mathcal{H}(\pi(\cdot|s_t))$ in the policy update to increase the policy entropy of the visited states. Thus, we define the reversed soft $Q$-function $Q_R^\pi(s_t, a_t)$ for MME as

$$Q_R^\pi(s_t, a_t) := r_t + \mathbb{E}_{\tau_{t+1}\sim\pi}\left[\sum_{l=t+1}^{\infty}\gamma^{l-t}(r_l - \alpha_Q\mathcal{H}(\pi(\cdot|s_l)))\right], \tag{7}$$

whereas the original soft $Q$-function of SAC in (2) is given by

$$Q^\pi(s_t, a_t) := r_t + \mathbb{E}_{\tau_{t+1}\sim\pi}\left[\sum_{l=t+1}^{\infty}\gamma^{l-t}(r_l + \alpha_\pi\mathcal{H}(\pi(\cdot|s_l)))\right].$$

Note that the original soft $Q$-function $Q^\pi$ adds the policy entropy to the reward and drives the policy to visit states with high entropy. On the other hand, our reversed soft $Q$-function subtracts the policy entropy from the reward and drives the policy to visit states with low entropy. In this sense, we call $Q_R$ as the "reversed" soft $Q$-function because it desires the reverse behavior of soft $Q$-function.

Then, $Q_R^\pi$ is estimated by a real-valued function $Q_R : \mathcal{S} \times \mathcal{A} \to \mathbb{R}$ based on a Bellman operator $\mathcal{T}_R^\pi$:
$$\mathcal{T}_R^\pi Q_R(s_t, a_t) = r_t + \gamma\mathbb{E}_{s_{t+1}\sim P(\cdot|s_t,a_t)}[V_R(s_{t+1})], \tag{8}$$
where $V_R(s_t) = \mathbb{E}_{a_t\sim\pi(\cdot|s_t)}[Q_R(s_t, a_t) + \alpha_Q\log\pi(a_t|s_t)]$ is the reversed soft state value function. At each iteration, the policy of MME is updated as
$$\pi_{new} = \arg\max_{\pi\in\Pi}\mathbb{E}_{a_t\sim\pi(\cdot|s_t)}[Q_R^{\pi_{old}}(s_t, a_t) - \alpha_\pi\log\pi(a_t|s_t)], \tag{9}$$

where $Q_R^{\pi_{old}}$ is substituted by the estimate function $Q_R$ at the iteration. Then, in pure exploration with no reward $r_t = 0, \forall t$, the policy of MME will visit the states with low entropy due to the first term $\mathbb{E}_{a_t \sim \pi(\cdot|s_t)}[Q_R^{\pi_{old}}(s_t, a_t)]$, and the policy entropy of those states will increase by the second term $\mathbb{E}_{a_t \sim \pi(\cdot|s_t)}[-\log \pi(a_t|s_t)] = \mathcal{H}(\pi(\cdot|s_t))$, as we intended. Note that the behaviour of the proposed method follows the max-min principle [11], so we expect that our MME fairly increase the policy entropy of all states based on the fairness perspective of max-min optimization, whereas SAC increases the policy entropy of states with high entropy only. The MME is expected to reduce the entropy difference and the $Q$-value difference between states to reduce the unwanted feedback loop and solve the saturation problem. Furthermore, SAC considers the same entropy coefficient $\alpha_\pi$ for its policy update and the soft $Q$-function $Q^\pi$, but our MME distinguishes the policy entropy coefficient $\alpha_\pi$ in the policy update (9) and the value-entropy coefficient $\alpha_Q$ in the reversed soft $Q$-function $Q_R^\pi$ in (7), as we experimented in Section 4. Changing $\alpha_Q$ and $\alpha_\pi$ allows for us to control the amount of the reversed $Q$-function in the policy update, and it will determine the ratio between the exploration due to the policy entropy and the exploration due to the reversed soft $Q$-function.

In actual implementation, the negative entropy in (7) is plus-offsetted to make the $Q$-update increase the $Q$-value. The detailed implementation and algorithm of MME are provided in Appendix A.

## 5.3 Disentangled Exploration and Exploitation for Rewarded Setup

In the previous subsection, we considered the problem from a pure exploration perspective. However, the ultimate goal of RL is to maximize the sum of rewards in rewarded environments, and the goal of exploration is to receive higher rewards without falling into local optima. With non-zero reward in (7) - (9), the policy will not only visit states with low entropy but also states with higher return. In this case, the reward and the entropy are intertwined in the $Q$-function and then it is difficult to expect the intended MME exploration behavior through the intertwined $Q$-function. Therefore, we disentangle exploration from exploitation for rewarded setup, as considered in several previous works [7, 49], and propose disentangled MME (DE-MME) for rewarded setup. For this, we consider two policies: pure exploration policy $\pi_E$ that samples actions for pure exploration as described in Sec. 5.2, and target policy $\pi_T$ that actually interacts with the environment. We decompose the reversed soft $Q$-function $Q_R^\pi$ in (7) into two terms $Q_R^\pi = Q_{R,R}^\pi + Q_{R,E}^\pi$, where $Q_{R,R}^\pi$ is the expected current and future reward sum considered in standard RL and $Q_{R,E}^\pi$ is the expected sum of future entropy:

$$Q_{R,R}^\pi(s_t, a_t) = r_t + \mathbb{E}_{\tau_{t+1} \sim \pi}\left[\sum_{l=t+1}^{\infty} \gamma^{l-t} r_l\right], Q_{R,E}^\pi(s_t, a_t) = -\alpha_Q \mathbb{E}_{\tau_{t+1} \sim \pi}\left[\sum_{l=t+1}^{\infty} \gamma^{l-t} \mathcal{H}(\pi(\cdot|s_t))\right].$$

Then, we update the policy $\pi_E$ for pure exploration as

$$\pi_{E,new} = \arg\max_{\pi' \in \Pi} \mathbb{E}_{a_t \sim \pi'(\cdot|s_t)}\left[Q_{R,E}^{\pi_{E,old}}(s_t, a_t) - \alpha_\pi \log \pi'(a_t|s_t)\right]. \tag{10}$$

Note that increasing the expectation of $Q_{R,E}^{\pi_{E,old}}$ makes the policy visit states with low entropy of $\pi_E$, as we intended in the pure exploration case in Sec. 5.2. Finally, we update the target policy $\pi_T$ by using $Q_{R,E}^{\pi_{E,old}}$ as

$$\pi_{T,new} = \arg\max_{\pi' \in \Pi} \mathbb{E}_{a_t \sim \pi'(\cdot|s_t)}\left[Q_{R,R}^{\pi_{T,old}}(s_t, a_t) + Q_{R,E}^{\pi_{E,old}}(s_t, a_t) - \alpha_\pi \log \pi'(a_t|s_t)\right]. \tag{11}$$

For implementation, $Q_{R,R}^{\pi_{T,old}}$ and $Q_{R,E}^{\pi_{E,old}}$ are estimated by real-valued functions $Q_{R,R}$ and $Q_{R,E}$ based on their own Bellman operators (see Appendix A). Note that the policy update (9) in Sec. 5.2 can be expressed as maximizing $\mathbb{E}_{a_t \sim \pi'(\cdot|s_t)}[Q_R^{\pi_{T,old}}(s_t, a_t) - \alpha_\pi \log \pi'(a_t|s_t)]$ over the target policy, where $Q_R^{\pi_{T,old}} = Q_{R,R}^{\pi_{T,old}} + Q_{R,E}^{\pi_{T,old}}$. Thus, we can view that the policy update in (11) replaces $Q_{R,E}^{\pi_{T,old}}$ in the previous policy update (9) with $Q_{R,E}^{\pi_{E,old}}$ to disentangle exploration from exploitation. In this way, the policy update (11) will simultaneously increase the expectation of $Q_{R,R}^{\pi_{T,old}}$ to maximize the reward sum, the expectation of $Q_{R,E}^{\pi_{E,old}}$ to visit states with low entropy, and the policy entropy for diverse action. The detailed implementation and algorithm for DE-MME are provided in Appendix A.

## 6 Experiments

We provide numerical results to show the performance of the proposed MME and DE-MME in pure exploration and various control tasks. We provide source code for the proposed method at

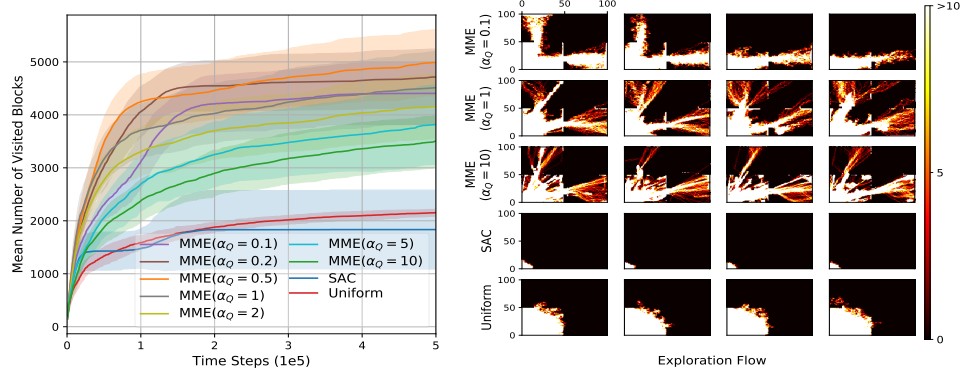

(a) Number of visited states     (b) State histogram of every 50k steps after 300k steps

Figure 3: Comparison of MME (Proposed), SAC, and the uniform policy in the 4-room maze

## 6.1 Pure Exploration

To see how the proposed method behaves in pure exploration, we considered the maze task described in Sec.4 again. We compared the exploration performance of MME in Sec.5.2, SAC, and the uniform policy. For MME, we considered several $\alpha_Q \in \{0.1, 0.2, 0.5, 1, 2, 5, 10\}$ with $\alpha_\pi = 1$. Fig. 3(a) in the next page shows the mean number of accumulated quantized visited states averaged over 30 random seeds corresponding to Fig. 1(b), and Fig. 3(b) shows the

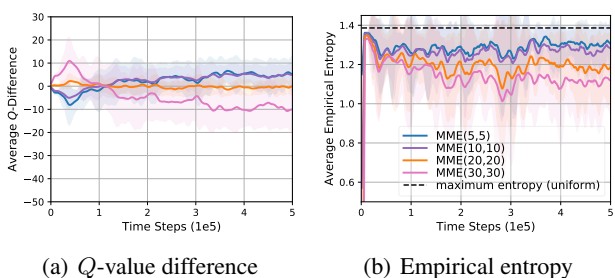

(a) $Q$-value difference     (b) Empirical entropy

Figure 4: Performance of MME

histogram of visited states, of which setup is the same as Fig.1(d). As seen in Fig. 3(a), the proposed MME visits much more states than SAC or the uniform policy. In addition, we observe that MME continues discovering new states throughout the learning, while SAC rarely visits new states as learning progresses. As seen in Fig. 3(b), MME explores far and rare states as compared to SAC or the uniform policy, and this leads to a large enhancement in exploration performance, as intended in Sec. 5.2. Note that the larger $\alpha_Q$ in update (7) (9) with $r_t = 0$, the stronger is the effect of the $Q$-function term to visit states with low entropy and the weaker is the effect of the policy entropy term to explore widely in the action space, as we expected in Section 5.2. Hence, there is a trade-off between the two terms and $\alpha_Q = 0.5$ seems best in the maze task when $\alpha_\pi = 1.0$, as seen in Fig. 3(a). Thus, the result clearly shows why we distinguish the policy entropy coefficient $\alpha_\pi$ and the value entropy coefficient $\alpha_Q$ for MME, whereas SAC uses the common entropy coefficient $\alpha = \alpha_\pi = \alpha_Q$. We also plotted the $Q$-value difference and the empirical entropy of the four squares centered at (5,5), (10,10), (20,20) and (30,30) for MME, as done in Figs. 2(b) and 2(c). The result is shown in Fig. 4. It is seen that the $Q$-value difference and the entropy difference among the states are clearly reduced as compared to Figs. 2(b) and 2(c). It means that MME breaks the unwanted positive feedback loop and improves the policy entropy of diverse states more uniformly as compared to SAC in terms of fairness under our max-min framework. This leads to better exploration, as seen in Fig 3.

## 6.2 Performance in Rewarded Environments

As mentioned in Sec.5.3, the ultimate goal of RL is to maximize the reward sum in rewarded environments and exploration is one of the means to achieve this goal. Based on the enhanced

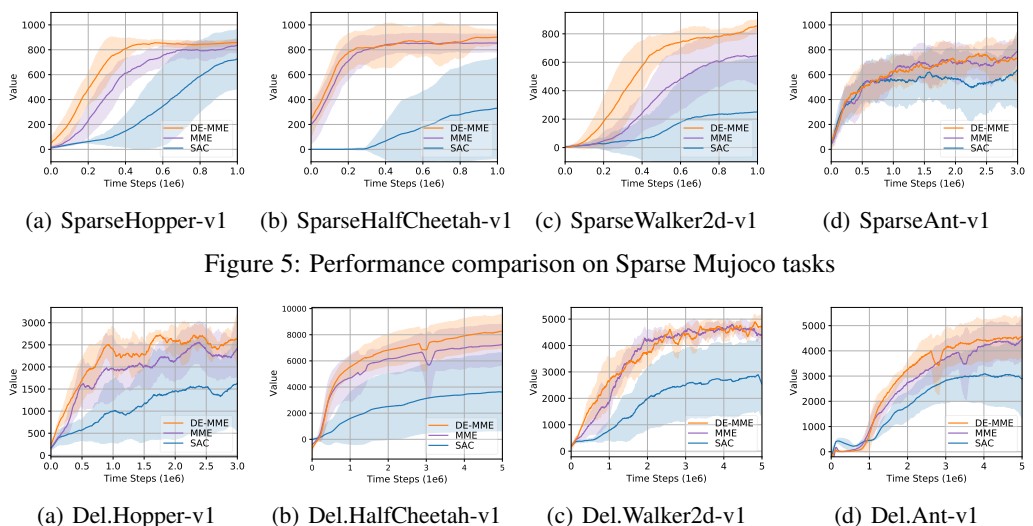

Figure 5: Performance comparison on Sparse Mujoco tasks

Figure 6: Performance comparison on Delayed Mujoco tasks

exploration performance of MME, we expect MME/DE-MME to show good performance in rewarded environments too. In order to verify this, we considered three types of difficult control tasks for which current state-of-the-art RL algorithms do not show satisfactory performance: Two types of sparse-reward tasks (SparseMujoco tasks and DelayedMujoco tasks) and high dimensional Humanoid tasks. SparseMujoco [27, 35] is a sparse version of Mujoco [52] in OpenAI Gym [8], and the reward is 1 if the agent crosses the $x$-axis threshold $\tau$, otherwise 0. DelayedMujoco [19, 57] is a delayed version of Mujoco in which the reward is accumulated for $D$ time steps and the agent receives the accumulated reward sum once every $D$ time steps. During the accumulation time, the agent receives no reward. These sparse-reward environments have widely been considered as challenging environments for validating the performance of exploration in many previous works [9, 25, 27].

First, we compared the performance of MME algorithms to that of maximum entropy SAC in the sparse-reward tasks. For MME, we considered two versions: vanilla MME proposed in Sec.5.2, and disentangled MME (DE-MME) proposed in Sec.5.3. For MME/DE-MME, we fixed $\alpha_\pi$ of MME and DE-MME to be equal to $\alpha$ of SAC, and chose proper $\alpha_Q$ for each task. Detailed experimental setup is provided in Appendix B. Figs. 5 and 6 show the performance averaged over 10 random seeds on SparseMujoco tasks and 5 random seeds on DelayedMujoco tasks, respectively. It is seen that the proposed MME shows much higher performance than SAC in the considered environments with rewards. It is also seen that MME itself performs well enough in most environments but DE-MME indeed yields performance gain over vanilla MME and the gain is large in SparseWalker. Thus, disentanglement of exploration from exploitation is beneficial to MME for better reward performance in rewarded environments, as discussed in Sec.5.3. We provided the corresponding max average return tables in Appendix C and ablation study for further analysis in Appendix D. There, one of ablation study empirically shows that the performance enhancement by MME is caused by improved exploration of MME as we intended.

Finally, we compared the performance of MME/DE-MME to that of popular general RL algorithms and recent exploration methods on the considered sparse-reward environments (SparseMujoco and DelayedMujoco tasks) and dense-reward high-dimensional Mujoco tasks (Humanoid, Humanoid-Standup). We considered several action-based exploration methods: SAC combined with divergence [27] (SAC-Div) and diversity actor-critic (DAC) [25], and state-based exploration methods with random network distillation (RND) [9] and MaxEnt (State) [26]. For general RL algorithms, we considered several on-policy RL algorithms: proximal policy optimization (PPO) [48] and trust-region policy optimization (TRPO) [47], and entropy-based off-policy RL algorithms: soft Q-learning (SQL) [21] and SAC [22]. We provided detailed explanation and implementation for each algorithm in Appendix C. Table 1 summarizes the max average return result. It is seen that MME/DE-MME have superior performance to other methods.

| | MME | DE-MME | DAC | SAC-Div | RND | MaxEnt(State) |
|---|---|---|---|---|---|---|
| Sps.Hopper | **902.50±4.36** | 893.30±6.72 | **900.30±3.93** | 817.40±253.54 | **897.90±6.06** | 879.50±30.96 |
| Sps.HalfCheetah | 903.50±34.97 | **924.90±39.57** | **915.90±50.71** | 394.70±405.53 | 827.80±85.61 | **924.70±24.44** |
| Sps.Walker2d | 818.00±208.60 | **886.60±25.77** | 665.10±355.66 | 278.50±398.23 | 750.90±179.09 | 705.30±274.88 |
| Sps.Ant | 953.70±28.39 | **973.60±12.55** | 935.80±37.08 | 870.70±121.14 | 920.60±107.50 | 940.70±43.84 |
| Del. Hopper | **3421.32±88.29** | **3435.28±39.55** | **3428.18±69.08** | 2090.64±1383.83 | 2721.06±1199.20 | 3254.10±30.75 |
| Del. HalfCheetah | 7299.28±1562.19 | **8451.20±1375.27** | 7594.70±1259.23 | 4080.67±3418.07 | 7429.94±1383.75 | 7907.98±535.41 |
| Del. Walker2d | 5148.58±193.78 | **5274.89±186.35** | 4067.11±257.81 | 4048.11±290.48 | 4098.63±683.36 | 4430.61±347.02 |
| Del. Ant | 4664.04±836.37 | **4851.64±830.88** | 4243.19±795.49 | 3978.34±1370.23 | 1361.36±704.69 | 1156.61±112.40 |
| | MME | DE-MME | SAC | SQL | PPO | TRPO |
| Humanoid-Standup | **267734.03 ±74302.99** | 250935.53 ±49386.43 | 167394.36 ±7291.99 | 138996.84 ±33903.03 | 160211.90 ±3268.37 | 153919.84 ±1575.62 |
| Humanoid | **9080.54±768.52** | 8607.75±570.61 | 6760.81±267.78 | 5010.72±248.59 | 6153.54±246.95 | 5730.74±455.90 |

Table 1: Max average return of MME/DE-MME and other recent RL algorithms

# 7 Conclusion

In this paper, we have proposed a MME framework for RL to resolve the unwanted exploration behavior of maximum entropy RL in off-policy learning with function approximation. In pure exploration, to implement MME, we train the $Q$-function to visit states with low entropy contrary to the maximum entropy strategy, while maintaining the policy entropy maximization term in the policy update. Then, we extended MME to rewarded environments. In rewarded environments we disentangle exploration from exploitation for MME to explore diverse states as in pure exploration as well as to achieve high return. Numerical results show that the proposed MME explores farther and wider in the state space than maximum entropy realization, alleviates possible positive feedback of off-policy maximum entropy learning, and yields a significant enhancement in exploration and final performance over existing RL methods in various difficult tasks. As for potential impacts, RL can be applied to sensitive areas that require control, such as drone control. However, it is only a risk that RL itself has, and it is not very relevant to the work that we are trying to address in this paper.

# 8 Acknowledgement

This work is supported by Center for Applied Research in Artificial Intelligence (CARAI) grant funded by Defense Acquisition Program Administration (DAPA) and Agency for Defense Development (ADD) (UD190031RD). Dr. Seungyul Han is currently with AI Graduate School of UNIST and his work is partly supported by Artificial Intelligence Graduate School support (UNIST), Institute of Information & Communications Technology Planning & Evaluation (IITP) grant funded by the Korea government (MSIT) (No.2020-0-01336).

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
