# OpenReview forum: "A Max-Min Entropy Framework for Reinforcement Learning"
_NeurIPS.cc/2021/Conference — NeurIPS 2021 Poster_

### Official Review · Reviewer_NuTp · 2021-07-15

**Rating:** 7
**Confidence:** 3

**Summary:**

This paper investigates the undesired behavior of soft actor-critic (SAC) algorithm that implementing maximum entropy reinforcement learning (RL).

One of the main contributions of this paper is the empirical investigation of the undesired behavior of SAC. By running SAC with function approximation on no-reward environment, the authors illustrate that the policy obtained by SAC failed to maximize the entropy, while maximizing the entropy is the optimal situation in no-reward environment and the theoretical study of SAC in finite MDP setups guarantees its optimal convergence. The authors pointed out a possible defect in SAC with function approximation: Q value of already visited states tend to be greater than the other states because the policy update increases the entropy of the visited states.

The second contribution is the proposal of a solution to the above mentioned defect. The authors propose Max-Min Entropy RL (MME). To improve the exploration, the soft Q-function has been revised. Instead of adding the entropy term to the Q-function, the revised soft Q-function subtracts the entropy, if I borrow the authors statement, to enhance the chance to visit rare states with low entropy. This is, however, counter intuitive to me as I will describe in the main review part.

The proposed approach has been tested on sparse mujoco tasks and delayed mujoco tasks, where the environments have sparse-reward shapes, as well as on some standard mujoco environments with dense rewards. The algorithm has been compared to different SOTA approaches. Promising performance has been observed.



**Limitations And Societal Impact:**

The limitation is the lack of theoretical guarantee, which the original SAC has. This is stated in Appendix. However, since the statement of the limitation is a very important part of a research paper that may promote a future investigation, I would like to suggest to put it in the main text.

**Main Review:**

The first contribution mentioned above is a very nice finding of a possible defect of SAC. It may open a new direction of future works. One small comment: This behavior of over-estimation of Q-value for already visited states may be related to the over-estimation bias of Q-value reported in TD3. The authors might want to comment on the similarity between them.

My concern is in the second contribution, i.e., the revised soft Q-function (7). This definition follows the authors idea of "requiring the policy to visit rare states with lower entropy more frequently." However, the rare states do not necessarily have lower entropy. Based on the authors empirical investigation on the defect of SAC, I understand that SAC tends to over-estimate the entropy of already visited states. Therefore, unvisited states have relatively smaller estimated value of the entropy. However, this is the matter of estimation, and the entropy itself is not necessarily lower for unvisited states. If we run the approach sufficiently long, Q-function is expected to tend to the right value, which is the right entropy value in case of no-reward situation. Therefore, I couldn't understand why this is the right modification to do. I appreciate if the authors give more justification of (7).

Although I am uncertain about the validness of the proposed modification, the empirical results presented in this paper are promising. One minor comment: What happens if the proposed approach is applied to swimmer environment? It is known to be deceptive and more exploration tends to lead to better performance, as is often shown in evolutionary RL and related approaches.


**Time Spent Reviewing:**

2

---

> ### Author Response · Authors · 2021-08-10
> **Reply to Reviewer NuTp**
>
> Q1) The first contribution mentioned above is a very nice finding of a possible defect of SAC. It may open a new direction of future works. One small comment: This behavior of over-estimation of Q-value for already visited states may be related to the over-estimation bias of Q-value reported in TD3. The authors might want to comment on the similarity between them.
>
> A1) Thanks for the insightful comment. We will investigate the impact of the over-estimation bias. If such a relationship is found, we think that we can modify the currently proposed algorithm by combining multiple Q estimation in addition to the max-min  idea.
>
> Q2) My concern is in the second contribution, i.e., the revised soft Q-function (7). This definition follows the authors idea of "requiring the policy to visit rare states with lower entropy more frequently." However, the rare states do not necessarily have lower entropy. Based on the authors empirical investigation on the defect of SAC, I understand that SAC tends to over-estimate the entropy of already visited states. Therefore, unvisited states have relatively smaller estimated value of the entropy. However, this is the matter of estimation, and the entropy itself is not necessarily lower for unvisited states. If we run the approach sufficiently long, Q-function is expected to tend to the right value, which is the right entropy value in case of no-reward situation. Therefore, I couldn't understand why this is the right modification to do. I appreciate if the authors give more justification of (7).
>
> A2) First, it seems that we need to distinguish prediction and control. In prediction, as we all know, the policy $\pi$ is given and we estimate the Q-function of the given policy $\pi$. In this case, if we apply the Bellman backup operator repeatedly, the estimated Q-function will converge to the true Q-function associated with the given policy $\pi$.  In this case, we can talk about the true or right entropy value of the policy $\pi$ at each state when convergence is reached.
>
> However, here we are doing a control problem, that is, we are trying to find the optimal policy through (soft) policy iteration. Hence, the policy is continuously evolving  at each time step through policy update and value function update.   So, in the case of control, when we say convergence, we should consider  $Q\rightarrow Q^*$, $\pi \rightarrow \pi^*$ and thus $H(\pi(\cdot|s)) \rightarrow H(\pi^*(\cdot|s)), \forall s \in \mathcal{S}$. Now, without the reward term in the Q-function, what is  the optimal policy $\pi^*$ of the policy iteration of SAC at least for finite MDPs with true expectation operation? Since it is proven that SAC policy iteration achieves the optimal solution of $\max_{\pi}J_{MaxEnt}(\pi)$ in finite MDPs with true expectation operation (not sample-based approximation), the optimal  policy  $\pi^*$ is the policy taking uniform action at every state.  Ideally, the entropy of $\pi^*(\cdot|s_1)$ is the same as that of $\pi^*(\cdot|s_2)$, where $s_1$ and $s_2$ are two different states, as the reviewer noticed.
>
> As in usual argument, in case of a finite MDP with every state visited infinitely often by a proper exploration mechanism so that sample-based approximation converges to the true expectation using distribution, such convergence may be achieved. But, in reality of a large (or continuous) state space with  sample-based learning, such convergence is not guaranteed. In fact, it can fail. We observed this by seeing that the performance of SAC in the continuous maze falls short of that of the uniform policy.
>
> In model-free sample-based learning, estimation based on samples inevitably kicks in. Q-function (including entropy) estimation and samples used in estimation cannot be separated, and the positive feedback and performance saturation occur in the SAC case. In fact, such positive feedback is often observed in other RL problems with on-line learning, which updates a very narrow focused region in the entire state (or state-action) space.
>
> Please note that the main cause of the positive feedback and the saturation problem in the model-free sample-based implementation of MaxEnt RL by SAC is 'the difference in the $Q$-values (due to entropy) among states' during the learning.  This is  explained in Section 4 and Section 5.1.  Note that in the soft Q-function of SAC (eq. (2)), without the reward terms, $Q^{\pi^*}(s,a)$ with the uniform policy for every state $\pi^*$ is constant across all $s$, i.e., no difference across $s$.  Our max-min formulation with equations (7) and (9) is an effective way to flatten out the difference in Q-values (only entropy no reward case). In fact, this flattening-out strategy is more relevant to and helps the convergence to the true Q values for all states, i.e., $Q^{\pi^*}(s,a)$ with the uniform policy for every state $\pi^*$. We can see the flattening in Fig. 4(a).  We think that our implementation is a better implementation of the maximum entropy principle than the SAC implementation. Please note that in tasks with rewards, we use disentanglement of exploration and exploitation so that the argument goes through.
>
> Q3) One minor comment: What happens if the proposed approach is applied to swimmer environment? It is known to be deceptive and more exploration tends to lead to better performance, as is often shown in evolutionary RL and related approaches.
>
> A3) We compared the performance of SAC and  proposed MME/DE-MME on Swimmer environment in the same way as in  the paper (5 random seeds, $\alpha_Q=\{0.1,0.2,0.5,1.0,2.0,5.0,10.0\}$, 1m time steps). From the maximum average return result (MME: $88.60\pm 27.09$, DE-MME: $89.57 \pm 10.49$, SAC: $49.48 \pm 1.84$), we observed that MME/DE-MME  has significantly higher performance compared to SAC in Swimmer environment. Especially, the performance of SAC is low in Swimmer environment, as shown in benchmarks for spinning up implementations (https://spinningup.openai.com/en/latest/spinningup/bench.html), and it may be due to the saturation problem.

---

> > ### Comment · Reviewer_NuTp · 2021-09-01
> > **Thanks**
> >
> > Dear authors,
> >
> > Thank you for your detailed response and additional experimental results. I am now more convinced.

---

### Official Review · Reviewer_P2Px · 2021-07-15

**Rating:** 6
**Confidence:** 2

**Summary:**

The authors describe a positive feedback loop, that arises in SAC-based implementation of the maximum entropy reinforcement learning and hinders efficient exploration. Q values of frequently visited states are updated more, because these states are stored more into the replay buffer. The policy update increases the entropy of the visited states, i.e. if a high entropy is rewarded in the policy the frequently visited states will be selected even more. In the paper, Max-Min Entropy Rl is proposed that is supposed to break the unwanted feedback loop. It considerably improved the empirical performance in common Maze tasks and Mujoco tasks.

**Ethical Concerns:**

-

**Limitations And Societal Impact:**

Societal Impact (last two sentences of the paper): I perceive the formulations "only a risk" and "not very relevant" slightly unlucky in the context of ethics, but contentwise I agree with the authors that the paper does not pose a risk beyond the general dual use problematic in RL.

**Main Review:**

Originality:
* as far as I know the described limitation was not studied before. The proposed method is new, too.

Quality:
* The paper does not contain a theoretical analysis, but the experiments are analyzed in quite a detail. I like that the authors plotted the magnitude of the Q-values and the entropy.

Clarity:
* In the first sentence of the abstract you write "the limitation of ...", but the abstract never says what this limitation is. If you can summarizes the positive feedback loop in one sentence, I would add that. Otherwise I would change "the limitation" to "a limitation".
* the described problem is rather a problem with the SAC implementation of the maximum entropy framework than with the maximum entropy framework itself. The problem is that the sampling based approximation in SAC results in a positive feedback loop, that keeps SAC from converging to the optimal solution of the maximum entropy objective $J_{MaxEnt}$. While this is described later on, reading the abstract and the introduction I misunderstood it as a limitation of the maximum entropy framework itself.
* The paper is well written in general. Just one suggestion for improvement of the reading flow: The caption of Section 4 mentions that Section 4 considers pure exploration only, but the text itself does not mention it. This made me stumble across the sentence starting in line 97, that only holds for pure exploration.

Significance:
* The method is widely applicable. However, I am wondering if the situation would be different in MDPs with stochastic transitions $P$. The described example and the experiments are all with deterministic transitions, right? But even if not, I would still consider it significant.

Further comments:
* Is it easy to see what $ J_{MaxMin}$ would look like?
* In Section 5.1, in line 164, it says: "For Fig. 2(b), every 1000 time steps, we sampled ...", but the x-axis label in Figure 2(b) is "Time Steps (1e5)".


**Time Spent Reviewing:**

6

---

> ### Author Response · Authors · 2021-08-10
> **Reply to  Reviewer P2Px**
>
> Q1) In the first sentence of the abstract you write "the limitation of ...", but the abstract never says what this limitation is. If you can summarizes the positive feedback loop in one sentence, I would add that. Otherwise I would change "the limitation" to "a limitation".
>
> A1) Thanks for the comment. We will revise the paper as suggested.
>
> Q2) The described problem is rather a problem with the SAC implementation of the maximum entropy framework than with the maximum entropy framework itself. The problem is that the sampling based approximation in SAC results in a positive feedback loop, that keeps SAC from converging to the optimal solution of the maximum entropy objective . While this is described later on, reading the abstract and the introduction I misunderstood it as a limitation of the maximum entropy framework itself.
>
> A2) Thanks for the comment. With a large scope, our max-min entropy algorithm can also be viewed as a maximum entropy algorithm since we are increasing the entropy anyway, although how to increase the entropy is different from SAC.  We will revise the paper properly as suggested.
>
> Q3) The paper is well written in general. Just one suggestion for improvement of the reading flow: The caption of Section 4 mentions that Section 4 considers pure exploration only, but the text itself does not mention it. This made me stumble across the sentence starting in line 97, that only holds for pure exploration.
>
> A3) Thanks for the comment. We will revise the paper as suggested.
>
> Q4) The method is widely applicable. However, I am wondering if the situation would be different in MDPs with stochastic transitions. The described example and the experiments are all with deterministic transitions, right? But even if not, I would still consider it significant.
>
> A4) We mainly consider the continuous maze task for pure exploration analysis and the sparse-reward Mujoco tasks for performance comparison. These considered environments are deterministic as the reviewer mentioned. However, we basically do not assume deterministic transitions. Even if  $P$ is stochastic,
> the stochasticity will not be uniformly random across the entire state space but the distribution of $s_{t+1}$ will be around some region. In such cases, the saturation problem will occur again and there will be  the $Q$-function difference among states according to the frequency of state visits for SAC.
> We think that the proposed method can be applied to mitigate the saturation problem in such cases. Following the reviewer's comment, we plan to test the max-min idea on such environments after we  find good examples for such stochastic environments.
>
> Q5) Is it easy to see what $J_{MaxMin}$ would look like?
>
> A5) As a matter of fact, we thought about this issue too. At this point, it was not easy to directly represent the proposed method by using a fixed objective function $J_{MaxMin}(\pi)$.
>
> An ultimate goal is that we define a common objective function $J_{MaxMin}(\pi)$, and prove that the policy of  MME converges to the optimal policy that maximizes $J_{MaxMin}(\pi)$ based on the reversed soft Q-function in eq (7) and the MME policy update in eq (9), as in many convergence proofs in RL.   We left this as a future work as described in the limitations section.
>
> We also plan to investigate the MME idea to robust RL because the max-min or min-max is a typical setup in robust learning. This is also a possible future research direction.
>
> Q6) In Section 5.1, in line 164, it says: "For Fig. 2(b), every 1000 time steps, we sampled ...", but the x-axis label in Figure 2(b) is "Time Steps (1e5)".
>
> A6) The episode length  is 1k time steps, so we draw points every 1k time steps in figure. ''Time Steps (1e5)'' means the time scale of number in x-axis: $0,1,2,3,4,5\rightarrow0,1\times10^5,2\times10^5,3\times10^5,4\times10^5,5\times10^5$.

---

> > ### Comment · Reviewer_P2Px · 2021-09-10
> > **Thank you**
> >
> > Thank you very much for the clarification. I lean towards acceptance.

---

### Official Review · Reviewer_rp8A · 2021-07-15

**Rating:** 4
**Confidence:** 4

**Summary:**

This paper proposes a conjecture that the agent should visit the state with low entropy and then do the exploration of this low entropy state with the maximum-entropy principle. To prove this conjecture, the author does a simple experiment on the 4-room maze and illustrates the empirical entropy of the action at different states. There is an interesting observation that the rarely visited states have low entropy maybe due to the function approximation error or off-policy learning explained by the author. Given this observation, the author proposes an algorithm which penalizes the reward with negative entropy （eq7) . Based on this modified Q function, the agent does the policy improvement as the original soft-actor-critic algorithm. The empirical result shows that the proposed algorithm outperforms the baselines such as SAC and many others.

**Limitations And Societal Impact:**

Yes

**Main Review:**

pros:
1. The authors provide a simple experiment to demonstrate their observations which leads to the easy understanding of the core idea.
2. The empirical performance of the proposed algorithm is strong.

cons:

1. The contribution of this paper is somewhat incremental, since the negative entropy can be thought a special case of the intrinsic reward.
2. In some experiment it seems true that the rarely visited state have low entropy. The author explained that in some heuristic way in page 5 and 6. Does this conclusion hold for most RL problems? I hope the author can provide more rigorous analysis, since it is the core of this paper.

**Time Spent Reviewing:**

3 hours

---

> ### Author Response · Authors · 2021-08-10
> **Reply to Reviewer rp8A**
>
> Q1) The contribution of this paper is somewhat incremental, since the negative entropy can be thought a special case of the intrinsic reward.
>
> A1) Thank you for the comment, but the authors do not agree with the reviewer on this point. Please note that using the negative entropy as the intrinsic reward is  a partial   interpretation of the proposed approach, not the main contribution of the paper. Also, simply using  the negative entropy as the intrinsic reward does not yield the proposed algorithm in the general case of tasks with rewards. The proposed algorithm is beyond this.
>
> To the authors' opinion, the main contribution of our paper is that we raised a new problem regarding the maximum entropy RL framework by investigating the phenomenon that the performance of SAC is saturated in actual sample-based learning with function approximation and does not converge to that of the optimal policy even in a simple pure exploration task, although the theory proved its convergence in finite MDPs. We identified  the cause of this saturation  based on experimental analysis, and came up with an idea opposite to the conventional maximum entropy idea to resolve this saturation problem. From the entropy-wise perspective, in the soft Q-function, we use the negation of the entropy and in policy update we  maximize the sum of the estimate soft Q-function and the current policy entropy. In this way, we drive the learning to visit low-future-entropy states and increase the entropy of these states.
>
> The maximum entropy framework and SAC are one of the prevalent approaches in the RL community.   The maximum entropy principle is applicable to many areas in nature and engineering due to the 2nd law of thermodynamics: Gaussian distribution under the finite second moment constraint, exponential families under the feature constraints, Burg's theorem, only to name a few. However, when it comes to RL, what good can one expect from the maximum entropy principle? Is the maximum entropy principle truly a right way to follow? The maximum entropy RL has long been investigated but a clear answer is not obtained yet. One of the biggest successes in the maximum entropy RL  is the SAC algorithm, which successfully combined the maximum entropy principle into the RL formulation, and proved its convergence under finite MDPs. Through our study presented in the paper, we now realize that SAC's implementation of the maximum entropy principle is not perfect and there exists a room for further improvement. So, to our opinion,  our study and this paper are important to the RL community, as seen from other reviewers' evaluation and comments.
>
> Second, please note that the proposed algorithm is not simply applying SAC with negative entropy as an additional intrinsic reward.  In pure exploration tasks with no reward, one can view like that. However, in general tasks with reward (which is the ultimate problem that we consider in RL), such a simple approach can jumble the reward and entropy in an unpredictable way and  the reward can disturb the exploration, since the sign of the (future) entropy in the soft Q-function and the sign of the (current) entropy are not the same in the max-min formulation.  To circumvent this difficulty, we adopted the disentanglement of exploration and exploitation, and produces a stable algorithm for general tasks with rewards for the proposed max-min entropy framework.  Pleas note that indeed the disentangled max-min entropy algorithm performs better in   most of the  sparse-rewarded tasks considered in  Section 6.2 of the paper.
>
> Q2) In some experiment it seems true that the rarely visited state have low entropy. The author explained that in some heuristic way in page 5 and 6. Does this conclusion hold for most RL problems? I hope the author can provide more rigorous analysis, since it is the core of this paper.
>
> A2) In the experiments in considered the paper, rare states have low entropy, but the statement may not hold for other complex environments, as the reviewer mentioned. Indeed, rare states may have high entropy.
>
> However, the most important thing to solve the positive-feedback-induced saturation problem is to reduce the difference in the $Q$-values among states, regardless of the state's rarity.
>
> As described in Sec. 5.1 and shown in Fig. 2, the initial $Q$-function difference induces the policy entropy difference among states, as seen in Fig. 2(c). This causes the saturation problem. In order to reduce the policy entropy difference, the agent should visit states with low entropy more frequently, and increase these state's entropy to reduce the entropy difference. Hence, we defined the reversed soft $Q$-function to visit states with low entropy and use it instead of the original soft $Q$-function. With such a motivation, we proposed the max-min entropy framework to   reduces the difference and successfully solved the saturation problem.  As a result of this max-min entropy strategy, the policy visits states more evenly and better exploration followed.
>
> But, if we think about why the saturation happens in the maximum entropy framework with sample-based learning, it is not difficult to understand why the max-min entropy framework yields better exploration.  Suppose that $Q(s,a)$ is flat over $\mathcal{S}\times \mathcal{A}$ initially. With the policy update equation eq. (5) with $s_t$ from the samples from the replay buffer, the policy will be updated so that the policy entropy of $s_t$ becomes high (if we neglect the reward part).  Then, with the $Q$-function formula eq. (2), the $Q$-value of the visited state becomes high. Then, in policy update eq. (5), the first $Q$ term kicks in and the policy will be learned to visit the high Q-value state. But, if we neglect the reward, the high Q-value state is previously-visited and it entropy is learned high. So, a positive feedback is constructed. In this process, the less-visited or un-visited states did not have chances for their entropy to be increased.  So, under policy update formula eq. (5), by setting the soft Q function to the sum of reward and negative entropy, we can flatten the entropy across states and this yields better exploration by breaking the positive feedback that makes the frequently-visited states be visited more.
>
> Since we apply disentanglement of exploration and exploitation, even in rewarded set up, our argument holds. This is our contribution.

---

> > ### Author Response · Authors · 2021-09-03
> > **Additional reply to Reviewer rp8A.**
> >
> > One more indicative example of the effectiveness of the proposed approach is the result of the swimmer envirionment. As seen in the paper, the proposed algorithm outperforms other baselines in all the considered Mujoco tasks in the paper. We cannot run test for all other tasks, but we ran the algorithm in the swimmer environment. As mentioned by Reviewre P2Px, the swimmer environment is  known to be deceptive and more exploration tends to lead to better performance.
> >
> > We compared the performance of SAC and the proposed MME/DE-MME on the swimmer environment in the same way as in the paper (5 random seeds, 1M time steps). From the maximum average return result (MME: 88.60+/- 27.09, DE-MME: 89.57+/-10.49, SAC: 49.48+/-1.84), it is observed that MME/DE-MME has significantly higher performance compared to SAC in the swimmer environment. Especially, the performance of SAC is low in the swimmer environment, as shown in benchmarks for spinning up implementations (https://spinningup.openai.com/en/latest/spinningup/bench.html). This results additionaly shows the enhanced exploration capability of the proposed approach.

---

### Official Review · Reviewer_M4pi · 2021-07-16

**Rating:** 7
**Confidence:** 4

**Summary:**

The paper addresses limitation in max-entropy RL frameworks. These frameworks usually add a regularization term to the reward function to increase policy entropy to encourage exploration, but the effect can be limited in states where the Q-function dominates the expression.
The authors propose a new objective that better encourages RL agents to visit rare states with low entropy, and show empirically that their method is able to outperform the baselines used for comparison.

**Ethical Concerns:**

No ethical concerns.

**Limitations And Societal Impact:**

Not much has been said about limitations, the authors could expand a bit on this point.
No negative societal impact.

**Main Review:**

I very much enjoyed reading this paper. The writing was clear, there was a nice flow of arguments, there was some interesting insights on commonly used MaxEnt RL frameworks, and a concise change to address the limitations found.

I think RL practicioners might find this paper useful, specially when dealing with sparse reward settings.

I only have a few suggestions that I think might help the reader.

- Line 98: why is the first term "the sum of future entropy"? It's the sum of Q, so wouldn it be more like the sum of "entropy-regularized reward"?

line 120-121: From the plot I do see that SAC behaves very differently than Uniform, however I don't know if you could definitely say that SAC visits fewer states than uniform, as the variance is huge and sometimes it visits a lot more states than uniform.

- Eq. 7 defines the new Q function for MME but the difference to Eq. 2 is very subtle, it took me a bit of scrolling back and forth to find the difference. It would be useful for the reader if as the new objective is derived either the previous one is re-stated on the same page, or the differences being explained as they are introduced.

- why is it called "reversed" soft Q-function?

- I'm a bit confused with the notation alpha_{Q}. Does this mean that the value of alpha_{Q} depends on the current Q value?

- Just for clarification, Q_{R,R} and Q_{R,E}, are two separate functions maintained by two separate approximators (neural nets), right?


- Typo line 47: (approximiate) -> (approximate)

**Time Spent Reviewing:**

5

---

> ### Author Response · Authors · 2021-08-10
> **Reply to Reviewer M4pi**
>
> Q1) Line 98: why is the first term "the sum of future entropy"? It's the sum of Q, so wouldn it be more like the sum of "entropy-regularized reward"?
>
> A1) As seen in eq. (2), $Q^\pi(s_t,a_t)$ contains the return $r_t + \gamma E_{\pi} [\sum_{l=t+1}^{\infty} \gamma^{l-t-t}r_l]$ and the future entropy $\gamma E_\pi[\sum_{l=t+1}^\infty \alpha \gamma^{l-t-t}\mathcal{H}(\pi(\cdot|s_l))]$. (We found that the $\gamma$ term is missing in front of the expectation in eq. (2), and will correct this.)
>
> The reviewer is right when we consider both the reward and the entropy. But, the sentence starts with "From the entropy perspective, ..." in Line 97. By this, we meant that "if we  consider only the entropy terms,..."
>
> We will revised the sentence to carry the meaning clearly.
>
> Q2) line 120-121: From the plot I do see that SAC behaves very differently than Uniform, however I don't know if you could definitely say that SAC visits fewer states than uniform, as the variance is huge and sometimes it visits a lot more states than uniform.
>
> A2) As the reviewer mentioned, SAC may be able to visit a lot more states than the  uniform policy for some seeds, depending on how many more states SAC visits in the beginning. However, the important point in Fig. 1(b) is 'the increment in visited states over time' on top of the number of visited states. As shown in Fig. 1(d), SAC ($\alpha_Q=1$) is saturated after 300k time steps and visits only a very narrow area in the state space of the maze environment, while the uniform policy explores widely and continues visiting new states. From this point of view, we stated that SAC visits fewer states than the uniform policy (on average, after the policy is saturated). We will rewrite it to reflect this well.
>
> Q3) Eq. 7 defines the new Q function for MME but the difference to Eq. 2 is very subtle, it took me a bit of scrolling back and forth to find the difference. It would be useful for the reader if as the new objective is derived either the previous one is re-stated on the same page, or the differences being explained as they are introduced.
>
> A3) As the reviewer mentioned, I will put the two expressions on the same page so that the difference is clearly visible.
>
> Q4) why is it called "reversed" soft Q-function?
>
> A4) The original soft Q-function of SAC adds the policy entropy to the reward and drives the policy to visit states with high entropy. On the other hand, our reversed soft Q-function subtracts the policy entropy from the reward and drives  the policy to visit states with low entropy. In this sense,  we named it the reversed soft Q-function because it desires the reverse behavior of soft Q-function.
>
> Q5) I'm a bit confused with the notation $\alpha_{Q}$. Does this mean that the value of $\alpha_{Q}$ depends on the current Q value?
>
> A5) $\alpha_Q$ is a hyper-parameter. In SAC, $\alpha$ appears in the soft Q-function (eq. (2) of the paper) and also appears in the policy-update objective function (eq. (5) of the paper). These two $\alpha$'s are the same in SAC. However, we set up a new formulation so that these two $\alpha$'s can be different. We use $\alpha_Q$ and $\alpha_\pi$ as follows: $\alpha_Q$ is the weighting factor between the reward sum and the future entropy in the soft Q-function (eq. (2)), whereas $\alpha_\pi$ is the weighting factor between the estimated Q-value and the current policy entropy in the policy-update objective function (eq. (5)). We consider that $\alpha_Q$ can be positive or negative so that in the soft Q-function, the entropy term can be added to or subtracted from the reward.  By having two independent $\alpha_Q$ and $\alpha_\pi$, we can consider various possible situations to investigate the impact of the entropy on policy learning.   For example, when $\alpha_Q=0$ and $\alpha_\pi=1$, the soft Q-function is nothing but the original Q-function, so at each time step, the policy is updated to maximize the return while maximizing the current policy entropy only. Original SAC belongs to the case of $\alpha_Q > 0$ and $\alpha_\pi > 0$.  So, it tries to maximize the sum of the current policy entropy and the estimate of the future entropy plus  future reward.  When  $\alpha_Q < 0$ and $\alpha_\pi > 0$, the entropy term is subtracted in the soft Q-function (in other words, the negation of the entropy is added to the reward), while the positive current policy entropy is maximized in the policy update. Hence, this situation becomes the max-min approach proposed in this paper and this situation is investigated intensively in our paper.  Note that in eq. (7) of our paper we defined the negation of the $\alpha_Q$ in the reply as $\alpha_Q$ to clearly show that the entropy term is subtracted. So, in eq. (7), $\alpha_Q >0$ due to the negative sign in front of the entropy term in eq. (7).
>
> Q6) Just for clarification, $Q_{R,R}$ and $Q_{R,E}$, are two separate functions maintained by two separate approximators (neural nets), right?
>
> A6) Yes, we used two separate neural networks for $Q_{R,R}$ and $Q_{R,E}$, as the reviewer mentioned.
>
> Q7) Typo line 47: (approximiate) $\rightarrow$ (approximate)
>
> A7) Thank you for pointing out. We will fix the typo.

---

### Author Response · Authors · 2021-08-10
**Common response**

We thank the reviewers for the valuable comments.  We raised and investigated the saturation problem of soft actor-critic.  Then, we proposed an extension of the maximum entropy framework by introducing a new max-min entropy framework to mitigate the saturation problem of soft actor-critic. We hope that our response below answers the reviewers’ questions.

---

### Decision · Program_Chairs · 2021-09-27

**Decision:**

Accept (Poster)

**Comment:**

This paper proposes an interesting and novel view on MaxEnt RL: It observes that  SAC may suffer from a feedback loop since the Q values of already visited states tend to be greater than that of other states because the policy update increase the entropy. This in turn encourages the policy to visit these states even more. This defect is confirmed in a no-reward environment and a method to address this is presented with encouraging experimental results. While there could have been more rigorous investigation into the claim "rarely visited state have low entropy", the overall contribution of the paper is appreciated by most of the reviewers and the interesting take on MaxEnt RL warrants acceptance.